# Expanding mental health services in low- and middle-income countries: A task-shifting framework for delivery of comprehensive, collaborative, and community-based care

community-based initiatives; global mental health; global mental health delivery; healthcare system; mental health

**Author for correspondence:**
Paul Bolton,
Email: PBolton@usaid.gov

Paul Bolton[1], Joyce West[2] 🔾, Claire Whitney[3], Mark J.D. Jordans[4] 🔾, Judith Bass[2] 🔾, Graham Thornicroft[5], Laura Murray[2], Leslie Snider[6], Julian Eaton[7], Pamela Y. Collins[8], Peter Ventevogel[9] 🔾, Stephanie Smith[10], Dan J. Stein[11], Inge Petersen[12] 🔾, Derrick Silove[13], Victor Ugo[14], John Mahoney[15], Rabih el Chammay[16], Carmen Contreras[17], Eddy Eustache[18], Phiona Koyiet[19], Esubalew Haile Wondimu[20], Nawaraj Upadhaya[21] and Giuseppe Raviola[22]

[1]United States Agency for International Development, Washington, DC, USA; [2]Department of Mental Health, Johns Hopkins University Bloomberg School of Public Health, Baltimore, MD, USA; [3]International Medical Corps, Santa Monica, CA, USA; [4]King's College London, Centre for Global Mental Health, Health Service and Population Research Department, Institute of Psychiatry, London, UK; [5]King's College London, Centre for Global Mental Health, Institute of Psychiatry, Psychology and Neuroscience, London, UK; [6]Peace in Practice, Amsterdam, The Netherlands; [7]CBM Global Disability Inclusion, London, UK; [8]Department of Psychiatry and Behavioral Sciences and Department of Global Health, UW Consortium for Global Mental Health and International Training and Education Center for Health (I-TECH), University of Washington, Seattle, WA, USA; [9]United Nations High Commissioner for Refugees, Public Health Section, Geneva, Switzerland; [10]Department of Psychiatry, Brigham and Women's Hospital, Boston, MA, USA; [11]SAMRC Unit on Risk & Resilience in Mental Disorders, Department of Psychiatry & Neuroscience Institute, University of Cape Town, Cape Town, South Africa; [12]University of KwaZulu-Natal, Durban, South Africa; [13]University of New South Wales, Sydney, NSW, Australia; [14]The MHPSS Collaborative, Copenhagen, Denmark; [15]Centre for Mental Health, University of Melbourne VCCC, School of Population and Global Health, Global and Cultural Mental Health Unit, Parkville, VIC, Australia; [16]National Mental Health Programme, Ministry of Public Health, Lebanese Government, Beirut, Lebanon; [17]Socios En Salud Sucursal (Partners In Health), Lima, Peru; [18]Zanmi Lasante (Partners In Health), Mirebalais, Haiti; [19]World Vision International, Nairobi, Kenya; [20]International Rescue Committee, New York, NY, USA; [21]HealthRight International, New York, NY, USA; and [22]Department of Global Health and Social Medicine, Harvard Medical School, Boston, MA, USA

## Abstract

This paper proposes a framework for comprehensive, collaborative, and community-based care (C4) for accessible mental health services in low-resource settings. Because mental health conditions have many causes, this framework includes social, public health, wellness and clinical services. It accommodates integration of stand-alone mental health programs with health and non-health community-based services. It addresses gaps in previous models including lack of community-based psychotherapeutic and social services, difficulty in addressing comorbidity of mental and physical conditions, and how workers interact with respect to referral and coordination of care. The framework is based on task-shifting of services to non-specialized workers. While the framework draws on the World Health Organization's Mental Health Gap Action Program and other global mental health models, there are important differences. The C4 Framework delineates types of workers based on their skills. Separate workers focus on: basic psychoeducation and information sharing; community-level, evidence-based psychotherapeutic counseling; and primary medical care and more advanced, specialized mental health services for more severe or complex cases. This paper is intended for individuals, organizations and governments interested in implementing mental health services. The primary aim is to provide a framework for the provision of widely accessible mental health care and services.

## Impact statement

This paper is intended to assist mental health services planners in understanding and budgeting the resource requirements for comprehensive mental health services, particularly in low- and middle-income countries (LMICs). Few declarative descriptions of such systems and their requirements currently exist, making this task difficult as the need for such guidance has become more widespread. As mental health has gained increased public health awareness globally, so has recognition that mental health services must become part of health and social services. Senior health and social service officials working in government and service organizations are

increasingly required to evaluate and approve new mental health services. This requires answers to the basic questions: "What will these services look like? What will be required?" In exploring these questions, the authors reviewed existing materials that consider the nature of mental health services. However, most deal with principles and design considerations without proposing a specific system or adequately describing service requirements. Those that do so are at least partly based on Western systems of care that place most treatment responsibility in clinics or hospitals, in the hands of professional health or mental health workers. In contrast, here we focus on community-based models of care (by which we mean care that is readily available to people in their own communities, with the services delivered in the community or close by) that are connected to secondary and tertiary care when necessary. We draw on the excellent principles and considerations from many existing sources, and our own collective experience, to propose a specific system of mental health services. We discuss these principles and considerations, and apply them to design a framework of services, describing the personnel requirements in terms of training, duties, and how the various elements interact. The intended impact is therefore a system design framework for future mental health services.

## Introduction

### Global mental health needs, services gaps and implications

Despite the high prevalence and human and economic burden of mental health conditions in all countries, most people do not have access to effective interventions (Degenhardt et al., 2017; Thornicroft et al., 2017; Alonso et al., 2018). The World Mental Health Surveys estimate rates of minimally adequate mental health services in low- and lower middle-income countries at 3.7% for persons with major depressive conditions (Thornicroft et al., 2017), 2.3% for anxiety conditions (Alonso et al., 2018), and 1.0% for substance use conditions, compared to 20.0%, 13.8% and 10.3%, respectively for high-income countries (HICs) (Degenhardt et al., 2017). As international conflicts and climate change continue to worsen, with displacement of populations and lifestyles, we expect the need for mental health and psychosocial support services (MHPSS) to increase.

The lack of adequate mental health care, especially in low- and middle-income countries (LMICs), is partly due to limited public funding, with median annual government expenditures for mental health in low-income countries (LICs) estimated to be 0.08 USD (eight cents) per capita compared to 0.37 USD in lower middle-income and 52.73 USD in HICs (World Health Organization, 2021a). Resources for children and adolescents are significantly less despite their comprising a large segment of the population in many countries and the importance of mental health issues at these stages of life. Instead, most public mental health funds in LMICs go to mental hospitals (80%) (World Health Organization, 2018b). Although the role of general primary healthcare staff in detection and management of mental health conditions has long been considered essential (Passmore, 1979), they provide little or no mental health services in LMICs. Our experience is that referral pathways to viable mental health services remain largely informal, with remoteness, cost and stigma the major barriers to care for most individuals and families.

The World Health Organization (WHO) has concluded that "The resources available to tackle the huge burden are insufficient, inequitably distributed and inefficiently used, so that a large majority of people with mental, neurological and substance use (MNS) conditions receive no care at all (World Health Organization, 2018f), despite evidence of the efficacy and effectiveness of psychosocial and pharmacological interventions" (World Health Organization, 2017). Data from LICs and LMICs indicate a median of only 1.4 and 3.8, respectively, mental health workers per 100,000 population, compared to 62.2 in HICs (World Health Organization, 2021a). The 2018 Lancet Commission on Global Mental Health and Sustainable Development concluded that "… access to mental health services remains very poor and fragmented for most people in the world. Although effective interventions exist and affordable methods for their delivery have been developed, the scale-up of quality mental health services has not happened in most countries" (Patel et al., 2018).

The lack of effective mental health and substance use services in LICs results in large social and economic losses (World Health Organization, 2018d, e, 2022). Lacking other options, individuals with severe mental health conditions may be restrained at home by well-meaning yet desperate family members, or at traditional or religious healing sites. They face barriers to attending school and finding employment, leading to "further marginalization, poor education and reduced employment opportunities" (World Health Organization, 2018d). Mental health conditions frequently lead individuals and families into poverty (Funk et al., 2010), yet services for individuals are rare, and family-based services even rarer. Homelessness and inappropriate incarceration are far more common among people living with mental health conditions than for the general population, worsening their marginalization and vulnerability (World Health Organization, 2013a).

While advocating for increased mental health funding, it is essential to make optimal use of existing resources. The WHO developed and launched mhGAP in 2008 to strengthen and scale up care of MNS conditions by healthcare workers, especially in LMICs (World Health Organization, 2008). The mhGAP initiative provides multiple resources (World Health Organization, 2015a, 2016b, 2018f, 2019) that build upon previous global mental health programming to bring mental health interventions by non-specialists to scale. WHO, in collaboration with the broader global mental health field, endorsed and refined complementary, evidence-based, low-intensity and scalable psychological interventions for use by lay counselors (World Health Organization, 2015b, 2016a,b, 2020a,b). WHO also recently released a self-help guide (World Health Organization, 2021b). WHO, the United Nations (UN) Children's Fund, the UN High Commissioner for Refugees and the UN Population Fund have developed the Minimum Service Package for Mental Health and Psychosocial Support in Humanitarian Settings, consisting of key activities, methods and tools focusing on Health, Education and Protection programming and based on explicit MHPSS standards (Inter-Agency Standing Committee (2022)). Altogether, these resources cover provision of care in routine services as well as support in emergencies. Preparedness (or disaster risk reduction), response and building back better can work together to strengthen systems in the long-term, and together, these resources provide a robust set of practical tools to begin to guide front-line providers in delivering mental health care and services in low-resource settings, and represent a significant advance for the field (World

Health Organization, 2013c, 2022; Inter-Agency Standing Committee, 2021).

This paper describes a framework for the delivery of comprehensive, collaborative, and community-based care (C4) around which these and other mental health resources and approaches can be organized. The framework is intended as a basis for full and accessible care and support for people living with mental health conditions in low-resource settings generally. The framework is suitable for organizing services for common mental health conditions (depression, anxiety, stress-related conditions, traumatic responses and substance use) and severe conditions (psychoses such as chronic schizophrenia, severe bipolar disorder, severe depression and other debilitating conditions). Because mental health conditions have a wide range of causes, including biological and environmental, this framework includes social, public health, wellness and clinical services. It accommodates integration of stand-alone mental health programs into health and non-health community-based services (e.g., education), and with primary physical health care and other clinical services, such as HIV, tuberculosis, cancer, noncommunicable diseases, maternal and child health.

This framework is aspirational. It is the result of the authors' collective global mental health experience with existing programs, and informed opinions on how best to address these issues in a single, unified framework. The framework has not yet been fully implemented. In most places it will require substantially increased funding of the mental health sector above current levels. Given the increased interest in mental health arising from COVID-19, and the corresponding advocacy by WHO and others for increased investment as necessary to meet mental health needs, we believe that it is time for a more comprehensive framework to provide practical guidance on the use of increased mental health spending. This is both to provide advice on the best use of these funds as well as to advocate that increased funding be made available.

This paper is intended for individuals, organizations and governments who are considering creating or expanding mental health services: policymakers and planners; funders; professional groups; service organizations; and other implementers. This includes decision makers without a technical background in mental health. Therefore, the paper focuses on resource and implementation issues, including the types of staff needed, training logistics, levels of effort, specific duties, and how workers interact as a team. As with most mental health frameworks and models, the primary aim is the feasible and efficient delivery of accessible mental health care and services to close the "know-do" gap.

In this paper "mental health conditions" not only refers to conditions that fulfill the criteria for mental disorder as defined by classification systems such as the ICD-11, but also states of emotional distress that cause suffering and functional impairment, and increase the risk of clinical mental disorders. Throughout this paper the main focus is on "community-based" services that are locally accessible. This includes services provided in clinics within a community. This is a shift from focusing spending on centralized psychiatric hospitals whom few can reach, to services that are much more accessible, less restrictive and therefore often more humane. It is in keeping with a rights-based approach to health care for all.

## Proposed Comprehensive, Collaborative, and Community-based Care (C4) Framework

The comprehensive, collaborative, and community-based care (C4) framework is illustrated in Figure 1, followed by descriptions of each element (boxes) and how they interact (arrows). A primary

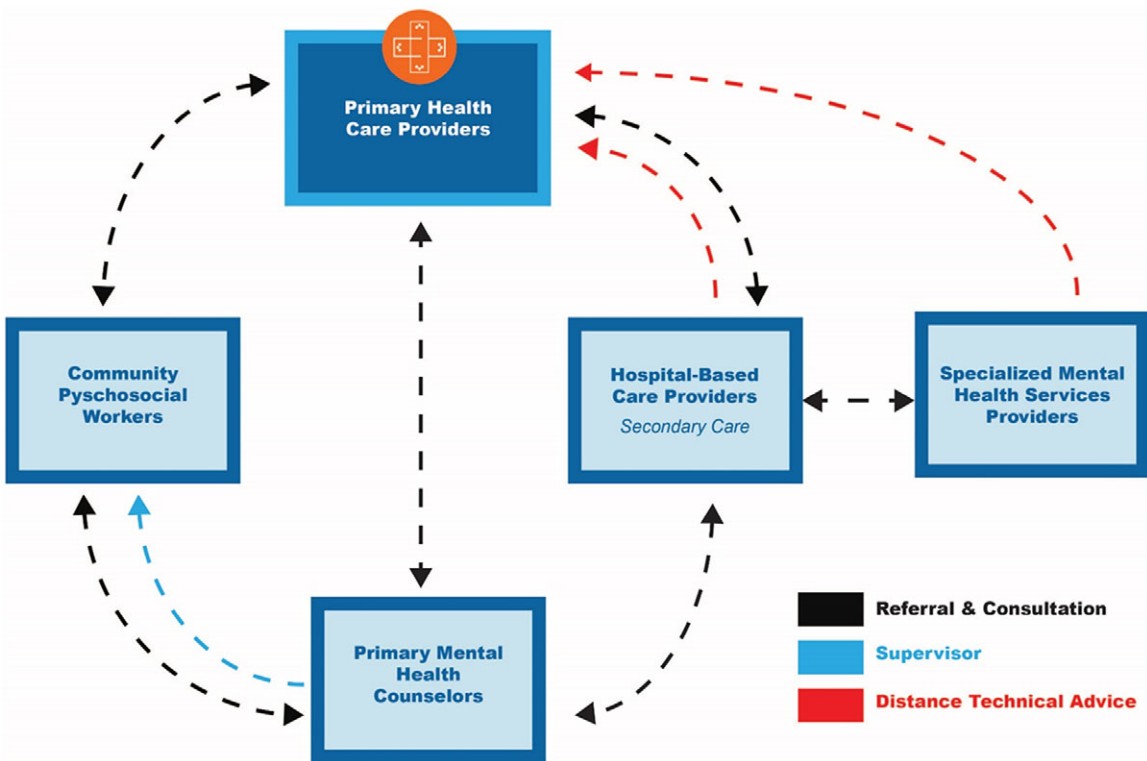

**Figure 1.** Comprehensive, collaborative, and community-based care (C4) framework for LMICs.

focus of the "C4 Framework" is to address the lack of accessible psychotherapeutic and social services. The C4 framework shares many features with the WHO's Transforming Mental Health for All services delivery model (2022). It also draws on the WHO's Mental Health Gap Action Program (mhGAP), and other global mental health models (summaries of these models are presented in Supplementary Material); but there are important differences.

First, the C4 Framework delineates types of workers based on their skills. Under the C4 Framework there are up to five categories of workers: Community Psychosocial Workers; Primary Health Care Providers; Primary Mental Health Counselors; Hospital-Based Care Providers; and Specialized Mental Health Services Providers (see Figure 1). These workers have separate but complementary roles, respectively: basic psychoeducation and information sharing; primary medical care; community-level, evidence-based psychotherapeutic counseling; and more advanced, specialized mental health services for more severe or complex cases. The framework emphasizes internal coordination, referral and back-referral mechanisms among service providers. Each staff has a complementary role to that of other staff, contributing to the overall team goal of delivery of comprehensive, collaborative mental health care in communities.

Second, the framework treats comorbidity of various mental health and substance use conditions as the norm, to be routinely treated by transdiagnostic counseling methods and other forms of support, care and treatment rather than separate, condition-specific interventions. Third, referrals are made outside of the community only for urgent pharmacologic intervention or inpatient safety monitoring. Fourth, the framework focuses specifically on family-based care. Mental health issues of one family member affect all family members. Fifty percent of mental disorders develop before the age of 14, and 75% before the age of 24 (Kessler et al., 2005). While children are often the focus of programming, the main mental health issues affecting them are often those of their parents. Similarly, women's salient mental health issues often include those of their male partners. Women bear a disproportionate burden in parenting and supporting the family unit, as well as caring for relatives with chronic mental disorders, notwithstanding the devastating effects to the family or mothers themselves developing or living with a mental disorder. A family-based approach promotes attention to this constellation of issues, whatever they are, as the best way of enhancing the mental health and well-being of any particular family member.

While five categories of workers may seem excessive, in low-resource environments a concerted focus on efficient use of resources requires more types of workers than high-resource environments. High-resource environments can generally afford to train and support professional health and specialty mental health providers who spend much of their time dealing with problems that do not tax their expertise. An example is primary care physicians in HICs who spend much of their time dealing with individuals who could just as well be treated by nurses or physician assistants. Without that luxury, low-resource environments need to maximize efficiency by training more types of specialized lay community workers, training the minimum number of each type of specialized workers required, and having these workers only see individuals who require their particular skills.

The point-by-point description below describes the required workforce and other resources to set up and maintain this framework. We do not recommend specific interventions as this would tie the framework to current knowledge. Instead, we refer to required attributes, particularly brevity, evidence-based,

acceptable, accessible, scalable and sustainable. Where specific interventions are mentioned, these are as examples.

This is a basic framework for low-resource environments. In environments with more resources the model could be different. For example, under the basic C4 Framework, Primary Health Care Providers do not refer directly to specialized mental health services. However, this could be included where specialized services are locally available. Where specialized services are locally available, Primary Health Care Providers could make direct referrals to them. Also, the tasks assigned to the workers described here could be taken on by different types of existing workers. More important is that the tasks are allocated to someone who is available, can be trained, and can provide sustained services.

## Community psychosocial workers

Community Psychosocial Workers have two roles: (1) directly promote mental health and well-being by providing prevention and coping interventions to people living and dealing with distress and stressful circumstances; and (2) identify and facilitate referral for persons needing care for mental health conditions. Community Psychosocial Workers need to be aware of the available services in their communities for people dealing with distress and stressful circumstances (e.g., legal, housing, protection), and how to provide referral information when needed.

### Functions of community psychosocial workers

1. Provide psychoeducation to the community, including basic information related to reactions to stress and regarding the nature of mental health conditions, to increase understanding of mental health. Reduce stigma. Assist individuals, including caregivers, in better handling life stressors. Increase appropriate demand for mental health and psychosocial services.
2. Teach basic coping skills and provide coping information for dealing with stressors, including supportive communication and practical support for people experiencing distress due to acute crises.
3. Provide information about non-health services, including social services and other support, and facilitate connections where needed.
4. Identify and refer individuals potentially in need of physical or mental health services to Primary Health Care Providers. Identify and refer to safety services those persons at risk of harm to self or others, experiencing acute psychosis or otherwise severely disabled, or in danger from others.
5. Conduct community visits, particularly checkups on individuals living with chronic and severe mental health conditions and their caregivers, targeting medication adherence, general self-care issues, and mobilization of social support resources. Refer back to Primary Health Care Providers where indicated.

Tasks 1–3 can be performed on a part-time basis through infrequent meetings (either in person or virtually) conducted weekly or less often, depending on the size of the population and the number of available workers. Individual assessments and referrals are the only tasks required between meetings. If there is sufficient time in Community Psychosocial Workers' schedules, two additional tasks are beneficial:

6. Foster social connectedness through group formation and facilitation, such as peer support groups. These activities can form part of community mobilization and citizen's

engagement for the general promotion of mental health and psychosocial well-being.

7. Liaise with key community stakeholders to help to address discrimination and neglect of people with mental health conditions, and to advocate for their rights and access to services and support. These activities, along with the community psychoeducation activities that include mental health conditions, are intended to increase understanding and support for persons with mental health conditions.

### Target population
Entire community.

### Location of services
Within the community, to enhance access and uptake of services. These services can be stand-alone, but are better integrated into existing public sector services by existing workers, both to enhance integration and to reduce costs by not requiring employment of a full-time worker. They can be part of the healthcare system, independent, or part of social, community or humanitarian services (e.g., education, religion, law enforcement or protection, justice system and social assistance). Services can be based in clinics or other service buildings, or at home. They can be provided to large or small groups, to individuals across the life course, and to families. They can include setting up support groups, organizing social and cultural events, or other acceptable approaches to building social support.

### Worker attributes
1. Established community members can provide these services part-time, and therefore this position and its duties can form part of a job providing other services, particularly those related to health, community rehabilitation, social services, protection or development. Better integration and reduced costs associated with not needing to hire other workers make combining these duties with an existing position more efficient. Individuals should be respected in the community, or at least acceptable to the local population, and be well-suited to providing management of community concerns related to health, or direct physical or emotional health support.
2. Whether working full-time or part-time, workers are adequately paid, particularly within the public sector, to give recognition to this role and to reduce turnover.
3. Preference to persons with lived experience of mental health conditions. Peer-based models are particularly effective in marginalized or stigmatized populations. This also counters the myth that persons with mental health conditions cannot hold jobs or take on socially valued responsibilities. Such workers appreciate mental health challenges and issues, and have an enhanced commitment.
4. Selection factors to be considered include gender, ethnicity and age, which vary by community. Good communication, negotiation and empathetic skills are also important.
5. A minimum of middle (primary) school education and ability to read and write in the local language. Comfort with use of mobile telephone devices is preferable.

### Training requirements
Approximately 3–7 days of didactic training with 2–3 practice sessions for each type of activity, followed by supervised practice for 5 or more program sessions until judged competent to conduct sessions independently. This is followed by weekly supervision until the supervisor confirms that the worker is fully competent, then less frequent supervision (bimonthly or monthly), plus ad hoc sessions as needed.

### Resources
1. Communications between provider and trained supervisor (see Section "Primary Mental Health Counselors" below) and with other workers. This includes support for their own well-being.
2. Meeting space for in-person sessions and/or telephone/internet for distance sessions.
3. Initial and refresher training sessions and materials, in electronic or paper formats.
4. Locally valid methods to identify potential mental health conditions and mental health crises (including suicidal ideation) and to facilitate appropriate referrals (Jordans et al., 2017).
5. Pay and reimbursement for providers for their transport and other costs.

## Primary Health Care Providers

Primary Health Care Providers are general healthcare providers working in non-specialized healthcare settings. They form the basis of healthcare systems in most countries. In HICs they are usually physicians, physician assistants or nurse practitioners. In many LMICs most Primary Health Care Providers are non-physicians with more limited primary care training. Most Primary Health Care Providers focus on physical health. Our experience is that LMIC Primary Health Care Providers are usually uncomfortable managing mental health conditions. This is due to existing workloads, lack of training, and the greater time required to treat these conditions. Also, psychological counseling skills are different from the typical advice-giving counseling skills of Primary Health Care Providers. Their training typically consists of history taking via closed questions followed by diagnosis and specific disease-focused treatments, requiring fewer visits. Under the C4 Framework we propose building on these existing strengths with management duties assigned to other workers with the requisite training. Primary Health Care Providers remain at the center of patient management, as they do for other health-related issues, since they are usually the most highly trained available care providers and the primary source of pharmacologic treatments. They are also best placed to deal with mental and physical comorbidity (World Health Organization, 2015a, 2018f). Integration of mental health services within primary health care is achievable in LMICs (Jordans et al., 2019b).

### Functions of Primary Health Care Providers
1. Receive and assess individuals, including referrals from other providers. WHO mhGAP provides algorithms suitable for Primary Health Care Providers to identify and manage a range of MNS disorders (World Health Organization 2016b). The management by Primary Health Care Providers involves developing a treatment plan, offering basic psychosocial support and psychoeducation, prescription of pharmacologic interventions when indicated, and referral to Primary Health Care Counselors, Specialist Mental Health Services Providers, hospital care or relevant social services and resources in the community. Assessment includes a medical evaluation to exclude physical health causes of psychological symptoms such as delirium, nutritional conditions and anemia.

2. Provide basic psychoeducation to increase awareness and knowledge, normalize individuals' experiences, reduce stigma, and promote intervention uptake and adherence.
3. Management of common and severe mental health conditions. This includes responding to psychiatric emergencies, and safety procedures and pharmacologic interventions where indicated, if possible in consultation with Hospital Based Care Providers or Specialized Mental Health Services Providers. If transfer to a hospital is indicated but not available, provide ongoing management within the community in consultation with Specialized Mental Health Services and/or Hospital-Based Care Providers.
4. Prescribe medications. For common mental health conditions, psychotherapeutic interventions are first-line treatments for individuals with mild to moderate symptoms. Treatment of depressive disorders or anxiety disorders often also involves medications, particularly when a combined approach is indicated or if other methods have failed. A wide range of generic medications are effective in treating mental health conditions, including severe psychotic and manic symptoms. Guidance is provided by the WHO mhGAP Intervention Guide (World Health Organization, 2015a, 2016b) and the WHO Guidelines for Management of Physical Health Conditions in Adults with Severe Mental Disorders (World Health Organization, 2018a). Primary Health Care Providers can initiate medications in the absence of a specialist provider. However, ongoing consultation and supervision from Specialized Mental Health Services Providers or Hospital-Based Care Providers is strongly recommended. Which medications are available at the PHC level will vary. Choice of medications should be consistent with the country's list of essential medicines, which should consider the WHO Model Lists of Essential Medicines (World Health Organization, 2021c).
5. Refer individuals to: Community Psychosocial Workers, for participation in group informational sessions and connection to other social services; Primary Mental Health Counselors, for full mental health assessment and psychotherapeutic interventions; and, where available, outpatient Specialized Mental Health Services Providers, for advanced assessment and pharmacologic treatment, or Hospital-Based Care Providers, for hospital admission and urgent care.
6. Receive back-referrals from Specialized Mental Health Services Providers and Hospital-Based Care Providers, and maintain and monitor medications and safety.

### Target population
1. Persons with mental health and psychosocial problems presenting directly or referred by community Psychosocial Workers or Primary Mental Health Counselors.
2. Individuals returning from treatment by Specialized Mental Health Services Providers or Hospital-Based Care Providers in need of follow-up care and/or monitoring.

### Location of services
Primary healthcare clinics within the community.

### Provider attributes
1. Trained Primary Health Care Providers within the formal healthcare system, working in or near the community.
2. Respected in the community or, at minimum, acceptable to the local population.

### Training requirements
Five to seven days of didactic training in assessment, de-escalation, management (including emotional crisis management) and referral or training in mhGAP-IG (or, in humanitarian settings, 3 days training in mhGAP-HIG). Training includes addressing biases in the management of persons with mental health conditions, such as diagnostic overshadowing. Didactic training should be followed by ongoing skills development through continuing medical education, supervision based on case reviews and client-based consultations with Hospital-Based Care Providers and Specialized Mental Health Services Providers.

### Resources
1. Communications between Primary Health Care Providers and other levels.
2. Local transport.
3. Meeting space for in-person sessions and telephone/internet for distance sessions.
4. Medications, both those prescribed by Primary Health Care Providers and those prescribed under specialist supervision.
5. Reimbursement for work and expenses, including cell phone and transportation.

## Primary Mental Health Counselors

Primary Mental Health Counselors focus on in-depth assessment for mental health conditions and provision of psychotherapeutic treatments where appropriate (Jordans et al., 2019a). In the C4 Framework they are primary care workers parallel to Primary Health Care Providers, providing community-based non-pharmacologic mental health services. They may be based in clinics or other service centers or provide services at home or other places accessible within the community and convenient for multiple visits. Ideally, they function as part of the formal, public sector health system. They provide interventions to individuals and small groups. They assess the socio-ecological environment of individuals seeking care and provide interventions for other family members as indicated.

### Functions of Primary Mental Health Counselors
1. Receive individuals seeking care directly or as referrals from Primary Health Care Providers and Community Psychosocial Workers, assess and provide treatment as indicated.
2. Develop and implement individualized and group intervention plans, including family-based interventions as indicated.
3. Act as case managers for those with mental health conditions. This includes monitoring psychotherapeutic treatments, medication compliance and treatment response, and adjusting plans accordingly.
4. Refer individuals, if indicated, for medications to Primary Health Care Providers, including those with severe mental health conditions such as psychosis, or with common mental health conditions that do not sufficiently improve with psychotherapeutic treatments.
5. Monitor safety – potential harm to self or others – and institute safety procedures in coordination with Primary Health Care Providers.
6. Supervise Community Psychosocial Workers.
7. Selected Primary Mental Health Counselors will also supervise the other Primary Mental Health Counselors (after additional training). They in turn will be supervised by the original trainers wherever possible, or a cadre of trainers developed over time.

### Target population

Individuals (children, youth and adults) referred from other providers or presenting directly with significant symptoms and associated dysfunction.

### Location of services

Community-based. Services are provided from a central site(s) that limits worker travel (maximizing time with individuals seeking services) and provides non-stigmatizing services other than mental health. Outreach services to families/homes, schools and other settings should also be provided as needed.

### Worker attributes

1. Established community members providing services full time or for most of their time. These services are too time consuming to be done part time.
2. Appropriately compensated as professional service providers according to local pay scales for equivalently trained and skilled providers. Training needs to be formally recognized.
3. Respected in the community or, at minimum, acceptable to the local population.
4. Consider gender, ethnicity, age and other factors important locally to ensure a balance that provides acceptable providers for all members of the population.
5. Where suitable candidates are available, preference would be given to persons with lived experience of mental health conditions. This would counter the widespread belief that persons with mental health conditions cannot hold jobs or take on responsibilities. It would also help to engage workers who are committed and clearly understand mental health issues.
6. Aptitude and interest in counseling, teaching and supervision.
7. Minimum of middle or primary school education and ability to read and write in the local language.

### Training requirements

Didactic instruction for 2–3 weeks, followed by weekly supervised practice with individuals receiving care (apprenticeship) for 6–12 months until competence is confirmed. Thereafter, regular monthly supervision, including regular competence and quality assurance assessments. Where more training is possible, it should be undertaken, including degree programs.

### Resources

1. Communications between Primary Mental Health Counselors and other levels, and with supervisor.
2. Local transport.
3. Meeting space for in-person confidential sessions and telephone/internet for distance sessions.
4. Reimbursement for expenses including cell phone and transportation.
5. Support for their well-being through the supervision process.

### Hospital-Based Care Providers (secondary care in general hospitals)

Hospital-Based Care Providers are physicians and nurses working in inpatient and outpatient secondary care facilities (local or district general hospitals) with access to typical hospital-based diagnostic and patient care services not available in primary care. In LMICs hospital-based mental health services are often limited to a few psychiatric hospitals in the capital or major cities, which is a model of care too often associated with human rights violations. Expanding the role of secondary care facilities to include outpatient psychiatric care and acute inpatient psychiatric care is essential to support efforts to integrate mental health into primary health care, as Primary Health Care Providers are unlikely to commit to mental health care delivery unless care, support and supervision are available at a higher level of the system. This requires psychiatric beds attended by staff trained in a consultation-liaison model of care (medical-psychiatric with additional attention to psychosomatic concerns), to move access to these services closer to communities. Where secondary hospitals do not have psychiatrists, medical and nursing staff can consult with specialists by phone or internet. Since most patients needing psychiatric care cannot get to specialized hospitals, this increases the typical quality of care by making psychiatric care more available and integrated into the health system.

This approach is required to reduce the use of restrictive environments where individuals are locked or chained, resulting in stigma and fear toward persons with severe mental health conditions. Referral from the community would be limited to those who temporarily require protection and psychiatric expertise for safety, stabilization and initiation of medications that cannot be provided locally. Once stabilized, individuals treated by Hospital-Based Care Providers would return to their communities for ongoing intervention and monitoring.

### Functions of general Hospital-Based Care Providers

1. Diagnose/confirm Primary Health Care Providers' mental and physical diagnoses, including ruling out physical causes of apparent mental health conditions.
2. Full-time acute inpatient monitoring and protection as required, using the least restrictive means of care. In some sites, this has included admitting a family member, particularly if they have traveled long distance. This provides an opportunity to educate them on the patient's care.
3. Stabilize and discharge patients with ongoing pharmacologic and counseling care and other psychosocial support as needed.
4. Manage discharge to the community, including communicating with Primary Health Care Providers with instructions for ongoing care.
5. Consult with Primary Health Care Providers and Primary Mental Health Counselors as needed, as they provide counseling and medication maintenance.
6. Consult with Specialized Mental Health Services Providers as needed, either in-house or at a distance.

### Target population

Individuals referred from other levels who are either a danger to self or others or who have not sufficiently improved with local care, and who have conditions that require access to psychiatric expertise in order to safely initiate treatment and stabilize persons in crisis.

### Location of services

Psychiatric inpatient facilities in general hospitals.

### Provider attributes

Hospital-based physicians and nurses with training in psychiatry and psychiatric nursing, and access to consultation with Specialized Mental Health Services Providers. In some cases psychiatrists and/or psychiatric nurses may be directly available.

### Training requirements

Three to seven days of didactic training, including stigma reduction, de-escalation and WHO QualityRights with a focus on recovery planning, advanced directives and supported decision making. This training would be followed by ongoing skills development through client-based consultations with Specialized Mental Health Services Providers.

### Resources

1. Hospital-based medications and testing facilities.
2. Psychiatric consultation and supervision of care, as needed.
3. Inpatient and outpatient care facilities.

## Specialized Mental Health Services Providers

Specialized Mental Health Services Providers include psychiatrists, psychologists and psychiatric nurses. They are not readily available to most LMIC populations, being few in number and usually working in large cities in private practice or psychiatric hospitals. In many countries psychologists do not receive significant clinical training and therefore are generally not able to provide evidence-based psychotherapeutic or psychosocial counseling. The C4 Framework therefore emphasizes community-based management of mental health and psychosocial problems, limiting reliance on Specialized Mental Health Services Providers' input to only when their expertise is needed. However, under this framework, expanded efforts to reduce stigma and improve identification and screening for problems at the community level will likely result in more referrals and more need for distance consultations. The demand for Specialized Mental Health Services Providers will likely increase in most LMICs, given their current limited supply. The need to improve quality at all levels and push services into communities reinforces the need for national guidelines to implement evidence-based strategies (i.e., in populations concurrently affected by common mental disorders and infectious diseases, or support for caregivers of people living with severe mental disorders).

### Functions of Specialized Mental Health Services Providers

1. Receive referrals from Hospital-Based Care Providers.
2. Consult on care by Primary and Hospital-Based Care Providers as needed.
3. Provide technical support to Primary Health Care Providers for urgent cases and persons with severe conditions who cannot leave their communities.
4. Where psychiatric hospitals exist, work with them to move away from long-term custodial care and focus on provision of care for people with the most complex psychiatric problems. Emphasize least restrictive means of care, short-term stays, community re-integration, and a family-based care system, to avoid unnecessary isolation from family and community supports.

### Target population

Psychiatric patients receiving care from Hospital-Based Care Providers who require specialized or advanced mental health services.

### Location of services

Hospital-based or distance consultation, depending on the service.

### Provider attributes and training

Providers include psychiatrists, psychiatric nurses and clinical or counseling psychologists or psychiatric social workers. Training should include WHO QualityRights and human right training, and introduction to community-based mental health services, evidence-based suicide assessment and safety planning, and evidence-based interventions.

### Implementation in settings with extremely limited specialist services

Where referral for severe conditions is not possible, remote support by specialists directly to local workers should be provided, with occasional visits by specialists to improve both local workers' practice, and the situational understanding of the specialist provider. Even in the absence of specialist expertise or medications, persons in crisis or with severe mental health conditions can benefit from the community-based providers under this framework. Their skills include empathetic listening and communication, psychoeducation, being supportive, giving hope, basic problem-solving, mobilizing social support, grounding techniques, relaxation and breathing exercises, behavioral activation, and simple cognitive-behavioral or interpersonal techniques. These can be delivered to individuals or to groups.

## Principles underlying the C4 Framework for LMICs

The C4 Framework is based on the key principles outlined below. These principles are based on our research and experience as well as the existing literature and work by the WHO, the Inter-Agency Standing Committee (IASC), and non-governmental organizations (NGO)s, including (but not limited) to those represented by the authors.

## Evidence-based practice

Mental health interventions must be based on scientific evidence and best practices (World Health Organization, 2013a). Scientific evidence consists of counterfactual trials in similar circumstances to proposed use. This is necessary in LMICs for interventions first developed in HICs.

## Collaborative, stepped care approach at all stages of mental health care delivery

The staging model recognizes opportunities for detection and intervention at all stages of mental health problems from stress to common mental health conditions to severe conditions (Berk et al., 2017). In the early stages, symptoms are often transient and not suggestive of a particular condition. Yet appropriate support and engagement can lead to better outcomes (McGorry and van Os, 2013) and are therefore preventive. Treating symptomatic persons early is important since common symptoms of mental distress such as anxiety symptoms or low mood are associated with more total population disability than clinical mental health conditions (Das-Munshi et al., 2008). The C4 Framework is designed to offer services with increasing intensity ranging from stress reactions to severe mental health conditions. It incorporates key components of collaborative care: a population-based approach to prevention including addressing psychosocial determinants of mental health and well-being; valid assessment and screening methods; brief psychotherapeutic interventions; access to mental health specialists for consultation and care supervision; and patient tracking (Kroenke and Unützer, 2017; Jackson-Triche et al., 2020; Unützer et al., 2020). Collaborative service delivery among providers should

be coordinated through case management by the Primary Mental Health Counselor, to ensure individuals get the care and resources they need without gaps or redundancy.

### Family-based care across the life cycle

To ensure effective community-based mental health and psychosocial support interventions for all members of the community, services must be tailored for individuals across the life course and designed to be inclusive of people of all genders, ages and abilities. This includes design and implementation of inclusive and developmentally appropriate services for infants, children, adolescents and adults of all ages, including the elderly, as well as for people with physical or mental disabilities. Consideration of the life course includes women who are pregnant, and perinatal mental health for mother and fetus. Comprehensively, this requires a "whole of family" approach that addresses the needs of people in the context of family resources and challenges, as well as potential additional training and supports to ensure adequate service delivery for special population groups. For example, adequate child and adolescent mental health services must consider how best to engage childcare providers, such as parents, extended family and teachers, and how to address the needs of out-of-school or homeless children as well as those with disabilities. How to tailor services to be developmentally appropriate and inclusive must be a part of training curricula and supervision at all levels of service delivery.

### Integration of MHPSS into community-based services

The need to integrate mental health services with physical health and other services (including protection, social and education) is well supported (Malone et al., 2007; Chatterjee et al., 2014; Dieterich et al., 2017; World Health Organization, 2022), and fundamental to the C4 Framework and other models. Some models propose counseling treatments be provided by Primary Health Care Providers. In many settings, however, Primary Health Care Providers are fully occupied with existing medical duties (Kok et al., 2017), while effective care requires multiple intervention sessions of an hour or more each. The mhGAP Operations Manual suggests providing mental health services through non-health platforms, including schools, neighborhoods, communities and workplaces, by teachers, police officers, social workers, traditional healers, services users, peers, parents, village elders and the general public (World Health Organization, 2018f). But at the same time considerations apply. Adding time-consuming psychotherapeutic services to their existing responsibilities makes it likely that these services will be a low priority, to be interrupted or set aside when other duties or urgent concerns intervene. Instead, it requires full time or "most of the time" commitment by dedicated providers. Other non-health providers can be trained in basic psychosocial support (such as psychological first aid) and referral. For example, teachers may benefit from learning how to better create safe and healing learning environments that aim to improve both mental health and psychosocial well-being, as well as learning outcomes, for children.

The C4 Framework splits community-based MHPSS into three types of elements: brief psychosocial interventions; evidence-based psychotherapeutic interventions; and pharmacologic interventions. Both the brief psychosocial and pharmacologic elements are integrated into the duties of existing workers. The psychosocial elements include group educational and skills building activities to be conducted several times a month or less frequently, to minimize the additional burden. These services are provided by the Community Psychosocial Workers which is preferably a part-time position for persons already engaged in social, educational or other community-based services as well as within, religious and cultural institutions that bring providers into contact with persons with MNS concerns. Integration can also occur within and across services for other co-occurring problems that risk complicating MNS concerns. This includes maternal health, nutrition, HIV, tuberculosis, early childhood development, child health and other conditions requiring behavioral change. Community Psychosocial Workers are therefore embedded to identify, assist and refer persons with MNS issues. More time-consuming psychotherapeutic treatments are provided by the separate group of dedicated Primary Mental Health Counselors.

### The central role of Primary Health Care Providers

According to the *WHO mhGAP Operations Manual*, "Caring for people with MNS conditions and chronic diseases must be provided in a person-centered, integrated approach, with integration at various levels, from screening and early detection of physical health conditions…and management of established physical and mental health conditions" (World Health Organization, 2018h). Some of the authors have previously promoted an integrated, patient-centered approach through greatly expanded training of Primary Health Care Providers in assessment and intervention (Thornicroft et al., 2019). The C4 Framework proposes expanding their training by building on their existing skills. Psychotherapeutic interventions, which use a very different skillset, are instead delivered by separate specialized providers with counseling-specific training. Primary Health Care Providers remain at the center of this framework, as they do for other physical and general medical care issues: receiving referrals; providing medication-based interventions; and consulting with and referring to other workers and providers as needed. This reduces the siloing of mental health from other general medical health services, which risks misdiagnosis and mismanagement. For example, persons with symptoms due to anemia and nutritional deficiencies can be misdiagnosed with depression when based solely on mental health assessments without a physical health assessment. It therefore helps to ensure that persons with mental health conditions receive adequate physical health care, and vice versa.

The C4 Framework expands the role of Primary Health Care Providers in dealing with severe conditions. Primary Health Care Providers not only assess and refer individuals with severe mental health conditions to hospitals, but also monitor and maintain treatment upon their return. This would provide for earlier discharge from hospitals and specialist care. This role is enhanced significantly by collaboration with Community Psychosocial Workers, facilitating better follow-up in the community.

### Dedicated local mental health workers treat multiple and comorbid conditions

Time-consuming care is provided by Primary Mental Health Counselors. As a new cadre of health workers, they require official recognition, including defined training curricula and requirements, pay scales and working conditions.

Since comorbidity is common, interventions should be transdiagnostic, using a single, flexible approach to address a variety of conditions. Workers learn a variety of intervention elements and

how to tailor varying element combinations to each individual to either treat a single problem or varying patterns of comorbidity (Murray et al., 2014). This is more feasible than training in multiple evidence-based psychotherapies that each focus on one or two conditions (e.g., Posttraumatic Stress Disorder (PTSD) and depression) (Kazdin and Blase, 2011), particularly in LMICs (Murray et al., 2014).

### Support of roles for people living with illness and families in clinical decision making

People living with illness and caregivers are typically not accustomed to providing input on their care and support (Lempp et al., 2018). Individuals with severe conditions are thought to have reduced capacity to make decisions. Families may not trust mental health services or fear using them due to stigma. Persons with MNS problems and their families have the right to make decisions regarding their own care in collaboration with mental healthcare providers, which can also improve services (World Health Organization, 2013b). Client input can make services more acceptable while family members can be instrumental in accessing important resources such as social support, recommended coping strategies, and self-help programs (Patel et al., 2018). The C4 Framework promotes client and caretaker involvement by education on the nature of mental health conditions and intervention options which can normalize mental health conditions and reduce stigma, and provide hope of effective management. Apprenticeship-type training (Murray et al., 2011) involves feedback from people living with illness, family and providers. Providers maintain communication with supervisors and supervisors with trainers/experts. The result is an ongoing four level conversation (person living with illness/caregivers, provider, supervisor and trainer/expert), producing iterative adaptations that improve care.

### Engaging persons with lived experience as providers

This increases the interest and understanding of mental health by providers even prior to training. It also enhances their credibility as mental health providers and makes them more approachable in the community. Engaging persons with lived experience as providers themselves also reduces stigma by challenging the common myth that they have reduced capacity to lead or take on responsible roles and more formal positions in their communities. The existing literature suggests that positive exposure to persons with lived experience is effective in reducing the stigma of mental health conditions.

### Workforce care, maintenance and development

Each type of worker requires supervision as well as training that includes attention to well-being, to maintain quality of care and reduce turnover. Community Psychosocial Workers receive a limited version of the training provided to Primary Mental Health Counselors. This enables Primary Mental Health Counselors to act as their trainers and supervisors, and to monitor and address the mental health of the Community Psychosocial Workers. Those Community Psychosocial Workers who are interested can be trained and become Primary Mental Health Counselors, providing professional development and more formal career advancement. Similarly, Primary Mental Health Counselors can become supervisors and ultimately trainers. The intent is to promote staff quality and retention by provision of a career pathway for workers at each level, and to reduce

staff turnover by selecting committed persons already within the framework for further training. This minimizes extra training and resource needs. Workers also learn from the extensive referral, back-referral and consultations built into the framework, which makes referrals, consultations and, ultimately, care, more efficient. This incorporation of professional development is missing from most service models.

With regard to supervision of providers by others with the same role, designated supervisors would have additional training and report to trainers. Community Psychosocial Workers would, however, be the exception to this and be supervised by Primary Mental Health Counselors who report to trainers.

Monitoring the well-being of workers as part of supervision and addressing their mental health needs are critical to sustainability. Mental health workers are subject to significant stresses and should not be harmed by their work. This represents an extension of "do no harm" and "self-care" principles to the mental health workforce.

### Severe mental health conditions must be addressed

Vulnerable to stigma and fear, persons living with severe mental health conditions may be abandoned or receive poor care, including acute and chronic physical restraint. These conditions include chronic schizophrenia, bipolar disorder, severe depression, severe substance use disorders or other debilitating conditions which can cause substantial disability. Individuals living with these problems are often the first to present to mental health programs. They are best initially assessed and managed by Primary Health Care Workers (who have received appropriate training and supervision on treating these conditions in line with the WHO mhGAP Intervention Guide or mgGAP Humanitarian Intervention Guide, for example) and require close monitoring and frequent follow-up. Consultation and, if possible, referral and stabilization under specialist care, may be needed for persons who do not respond to treatment, who experience serious side effects with pharmacological interventions, who have comorbid physical or MNS conditions, or who are at acute risk for suicide or self-harm. If in-person care is not possible, these individuals must be supported at home with support from Community Psychosocial Workers and Primary Mental Health Counselors, under specialist advice by phone or internet. The WHO QualityRights Initiative and Toolkit are key resources in assisting these populations in ways that protect human rights and are community-based as well as rehabilitation- and recovery-oriented (World Health Organization, 2012).

### Intellectual disabilities and developmental disorders must be supported

While specific treatments for intellectual disabilities and developmental disorders may not be available, those affected as well as their families and caregivers can benefit from the psychoeducation, training for caregivers, psychosocial support and referral (to other services when needed) offered by Community Psychosocial Workers. This includes identifying and treating associated mental health conditions that would not otherwise be recognized.

### Implementation issues and challenges

A complete review of all potential implementation issues and challenges that have been described in the literature is beyond the scope of this paper. The provider types and roles described above

need to be tailored to local resources, barriers and contexts (e.g., emergency, peace or development contexts). Local circumstances may sometimes dictate moving some of the described tasks to different types of providers than those presented in this C4 Framework, which would in turn affect the necessary training requirements and resources.

Most mental health problems can be addressed by locally trained individuals with sufficient supervision, and generally without requiring pharmacology or equipment. What is effective and acceptable across populations, how to train each category of worker, and how to connect them and support them, have been explored in previous work. The major challenge is the lack of funding for mental health services commensurate with their importance in addressing mortality, morbidity and disability. A rational approach to funding of public services, including health services, would transform the availability of effective mental health services. As community-based and local hospital psychiatric services become accessible, this would also enable a move away from custodial care in psychiatric hospitals, reducing the number and length of admissions, and increasing discharges of long-term residents.

In the following sections we address in more depth some major implementation issues and challenges that have emerged during the development of this approach and this paper.

### This approach will require a large increase in spending, both to set up and to maintain, which is currently not available in many LMICs

The WHO has recognized that substantial increases in global access to mental health services require a substantial increase in funding, and have consistently called for LMICs to substantially increase their funding for mental health services. Therefore, the requirement for greater resources is not specific to this framework. Resource limitations were a major consideration in the structure of the proposed framework. The authors believe that this approach would be a much lower cost option to universal access than simply calling for an increase in the number of psychiatrists, psychologists and other more specialized mental health staff employed in Western models of care.

Some work has been carried out to assess the costs associated with delivering a basic package of mental health care, based on similar approaches to those proposed in this paper; that is, based on decentralization, task sharing, and provision of basic medical and psychological interventions for major mental health conditions in primary care. Some of this work has been based on pilot programs, and some using modeling. In Nigeria, a package of care for psychosis, depression, anxiety, alcohol use and epilepsy was estimated to produce one extra year of healthy life at a cost of less than US $320, which is the Nigerian average per capita income (Chisholm et al., 2016). A return on investment analysis of services for depression and anxiety, using approaches similar to those included in the framework in LICs, estimated that for every USD 1 invested in mental health services, there was a return of US $3-5 (Gureje et al., 2007). This was deemed to be a conservative figure, as treatment outcomes were restricted to economic impacts at the individual and family levels. Mental health services have also been incorporated into the OneHealth Tool for costing and national health strategic planning in LMICs (Chisholm et al., 2017). Such work has led to the development of a methodology for supporting national investment decisions in mental health ("Investment Cases") (World Health Organization, 2021d).

Even with evidence of substantial cost-benefits for mental health, it is clear that these considerations are only one part of the equation to mobilize resources for mental health services reform. In LMICs, investments in mental health services have historically been pitifully low compared to the very high relative burden of diseases, with the limited mental health budgets tending to be inefficiently allocated, and with estimates that many countries with unreformed services spend over 80% of their budgets on outdated specialist hospitals (Saxena et al., 2007). Reallocation of resources toward more efficient, decentralized, community-based services has long been accepted as a logical approach, but political considerations often prevent reduction in funding to established services. Ideally, community-based alternatives need to be in place in advance of reform or closure of existing institutions; hence the need for "bridging funds" and clear commitments to reinvest funds saved from specialist hospitals to community-based services (World Health Organization, 2014).

### This approach assumes that the health system will have the willingness and human resources at specialist hospitals for implementation

In many LMICs both general and specialty hospitals do not routinely have adequate services of psychiatrists, psychologists, social workers or mental health nurses to carry out the designated roles outlined in this paper. In recognition of this challenge we stress that this framework is aspirational, suggesting what we are building toward as we advocate for more resources for mental health, while recognizing that not every locality or country currently has the required resources. Regarding resources at specialist hospitals, this proposed approach has already been successfully implemented in some countries. For example, in Sri Lanka after the 2004 tsunami, there were no services. Psychiatrists typically covered populations of a million people or more. Other than a few inpatient units, there were no staff trained in psychiatry so a model of full-time Medical Officers of Mental Health and non-professional Community Support Officers was implemented and proved to be highly effective (Mahoney et al., 2006).

### The framework assumes a high level of local leadership and governance mechanisms for the progressive implementation, refinement, local adaptation and scaling up of care

This is most often not the case. While high levels of local leadership and governance are often lacking in some places, they are present in others. The primary goal of this framework is to offer an approach for those places where there are effective leaders who are interested in seeking guidance to improve access to mental health services. They do not necessarily need to be knowledgeable about mental health. For example, in Sri Lanka service development was successfully led by the Regional Directors of Health, none of whom had any prior experience in providing community-based mental health services (Mahoney et al., 2006; Saraceno et al., 2007; Kakuma et al., 2011).

### While the demand for mental health services worldwide has increased considerably, the COVID pandemic has disrupted the already low levels of in-person mental health services

These challenges enhance the need for efficient mental health services through task-shifting to non-professional providers, better access by training local workers, and use of telemedicine,

teletherapy and other innovative virtual services or social media applications. Mobile phones, which are increasingly available worldwide, have been used to facilitate access to training, supervision, consultation and support among care providers and to make records available remotely. However, low digital literacy, low smartphone penetration in some places, digital data costs, and limited internet connection continue to make online mental health services a limited option in some locations.

## Summary and conclusion

The C4 Framework fulfills the rights of individuals to access effective mental health care according to need. Access includes facilitating appropriate care seeking by reducing stigma, and promoting social inclusion and empowerment of individuals living with mental health conditions. The framework has three levels of MHPSS: (1) brief psychosocial intervention delivered by Community Psychosocial Workers; (2) evidence-based psychotherapeutic interventions delivered by mental health counselors; and (3) pharmacological interventions delivered by trained physicians, nurses or psychiatrists. These can be expanded by: (a) training and supervising several types of workers with complementary roles; (b) establishing community-based identification and referral mechanisms; (c) mobilizing people with lived experiences to provide community-based services including an anti-stigma role; (d) integrating MHPSS considerations into community-based health, protection and social programs; (e) ensuring the treatment of comorbid conditions; (f) establishing home-based care approaches to support the family as a whole to care for a family member living with a mental health condition; and (g) instituting a community-based post-treatment follow-up mechanism.

Useful models, frameworks and innovations exist to improve access and quality of mental health and substance use services in low-resource settings. However, the materials describing this work often focus on general principles and can be vague in terms of worker roles and needed resources. The C4 Framework builds upon these existing models to propose a pragmatic, task-shifting approach that describes a specific structure of a comprehensive, collaborative, and community-based mental health system with focus on worker roles, training and resources. The intent is to make this framework accessible to not only global mental health specialists but also non-experts who increasingly need to be engaged in the development of mental health services, including governments, funding organizations, NGOs and other social and health services providers.

**Open peer review.** To view the open peer review materials for this article, please visit http://doi.org/10.1017/gmh.2023.5.

**Supplementary material.** To view supplementary material for this article, please visit https://doi.org/10.1017/gmh.2023.5.

**Acknowledgements.** The authors thank Inka Weissbecker and Mark van Ommeren for valued technical guidance, and Sama El Baz for critical contributions in finalizing this manuscript. The authors recognize the mental health teams of Inshuti Mu Buzima (Rwanda), Socios En Salud (Peru) and Zanmi Lasante (Haiti), as well as many other community-based organizations advancing mental health care delivery around the world.

**Author contributions.** All authors have made substantial contributions to the conception or design of this work and drafting or revising it critically for important intellectual content, and have approved the final version to be published.

**Financial support.** This work was supported by the U.S. Agency for International Development (P.B.); the National Institute of Mental Health (P.Y.C., P30 MH123248(Simoni); the National Institute for Health and Care Research (NIHR) Applied Research Collaboration South London (NIHR ARC South London) at King's College Hospital NHS Foundation Trust (G.T.); UK Medical Research Council (G.T., UKRI); Emilia (G.T., MR/S001255/1) and Indigo Partnership (G.T. MR/R023697/1) awards; the South African Medical Research Council (D.J.S.); Grand Challenges Canada (S.S., C.C., E.E., G.R.); and the Many Voices Foundation innovations grant to support strengthening of cross-national mental health programs at Partners In Health (S.S., C.C., E.E., G.R.).

**Competing interest.** Dr. Dan J Stein has received research grants and/or consultancy honoraria from Discovery Vitality, Johnson & Johnson, Kanna, Lundbeck, Orion, Sanofi, Servier, Takeda and Vistagen. The authors alone are responsible for the views expressed in this article and they do not necessarily represent the views, decisions or policies of the institutions with which they are affiliated.

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
