## [Reviewer Report]

October 26, 2022

Dear Dixon,

On behalf of my distinguished international colleagues and co-authors, we appreciate your consideration of our revised White Paper, “Expanding Mental Health Services in Low- and Middle-Income Countries: A Task-Shifting Framework for Delivery of Comprehensive, Collaborative Community-Based Care.” The goal of this White Paper is to describe an aspirational framework for Comprehensive, Collaborative Community-based Care (C4) for delivering full and accessible mental health services in low resource settings. It is the result of deliberations between the authors based on their implementation experience and their knowledge of the global mental health literature. 

This work is intended for consideration by individuals, organizations and governments who are considering creating or expanding mental health services. The proposed framework addresses gaps in previous models, including the lack of accessible community-based psychotherapeutic and social services, difficulty in addressing comorbidity of mental and physical conditions, and describing interactions between different types of workers in referral and coordination of care. 

The paper includes descriptions of the framework and underlying principles, but also focuses on what implementing this approach would require in terms of staff and resources. Because of this expanded description we are requesting that you waive the 4,500 word limit. The total word count is 9,248.

Thank you again for considering this work for publication in Global Mental Health.

Sincerely, 

Paul Bolton, MB BS MPH MSc

Senior Scientist, Department of Mental Health

Johns Hopkins University Bloomberg School of Public Health

Mental Health and Psychosocial Support Coordinator, USAID

---

## [Reviewer Report]

*Comments to Author*: Thank you for inviting me to comment on this paper, which introduces an innovative framework for Comprehensive, Collaborative Community-based Care (C4) for

accessible mental health services in low-resource settings. It is based on a task-shifting approach, that is robust, all-encompassing, and pragmatic while retaining flexibility for adaptation to the perculiar needs of different settings and available resources.

The C4 framework reflects the cumulative global mental health expertise of the authors, as it successfully attempts to improve on the limitations of previous interventional models in a holistic manner.

The authors admit that the framework is aspirational and will require improved funding in order for it to be implemented. But the mere existence of this framework is sufficient evidence to make a case for improved funding. The principles upon which the framework derives are outlined and the potential implementation challenges are duly identified and highlighted, along with potential solutions.

Overall, the C4 framework provides an exciting blueprint for a cost-effective, evidence-based and holistic approach to reducing treatment gap in low and middle income settings by expanding access to quality mental health care services. It takes on board, involvement of family members, respect and protection of human rights and a community-oriented approach that is more likely to promote acceptance.

All that remains is for me to congratulate the authors, as I look forward to evaluating the impact of the C4 framework on improved mental health services in low and middle income countries in the years to come.

---

## [Reviewer Report]

*Comments to Author*: To start, I should note that I was requested to review this manuscript following its re-submission to the journal; I have not seen the original manuscript but have read the initial reviewer comments and the authors' responses to them. My comments below take into account as much as possible this initial submission and review process.

1. Relevance: The paper addresses a highly relevant set of inter-related issues around the development and improvement of community-based mental health services in low- and middle-income country settings (LMIC). It seeks to confront and address known challenges and bottlenecks in moving from policy to practice, particularly with respect to service organization and human resource development and deployment.

2. Authorship: The paper and the C4 framework it proposes has been put together by a highly reputable group of global mental health researchers and champions, with very substantial experience of mental health services research and development in LMIC. However, it is noted that only a small minority of the authors actually come from / live in the LMIC settings that are the subject of the paper, which may raise legitimate concerns or criticism that it is not sufficiently grounded in the realities of those countries. However, I note that many of the initial reviewers' comments were expressing such concerns and the authors have taken appropriate steps to address them, for example by re-casting the proposed service network as 'aspirational'.

3. Framework: The cited reference frameworks in the Annex are all appropriate but there seems to be a major missing one, namely the comprehensive service framework set out in WHO's World Mental Health Report 2022. Probably this is because the paper was drafted before the publication of this Report in June of this year, but nevertheless it seems out of date to not include - and be influenced by - this recent articulation of how a comprehensive network of mental health services could look like, including in LMIC settings.

---

## [Reviewer Report]

February 8, 2022

Dear Dixon,

On behalf of my distinguished international colleagues and co-authors, we are grateful to you for inviting us to submit a revision of our manuscript, “Expanding Mental Health Services in Low- and Middle-Income Countries: A Task-Shifting Framework for Delivery of Comprehensive, Collaborative Community-Based Care.” 

We would be very pleased to have this work published in Global Mental Health to help advance improvements in mental health services throughout the world. 

Sincerely, 

Paul Bolton, MB, BS

Mental Health and Psychosocial Coordinator,

United States Agency for International Development